# Resistance to TOP-1 Inhibitors: Good Old Drugs Still Can Surprise Us

**DOI:** 10.3390/ijms24087233

**Published:** 2023-04-13

**Authors:** Santosh Kumar, Michael Y. Sherman

**Affiliations:** Department of Molecular Biology, Ariel University, Ariel 40700, Israel; kumars@ariel.ac.il

**Keywords:** irinotecan, drug resistance, topoisomerase I, dose escalation, chemotherapy, DNA repair, DNA damage response

## Abstract

Irinotecan (SN-38) is a potent and broad-spectrum anticancer drug that targets DNA topoisomerase I (Top1). It exerts its cytotoxic effects by binding to the Top1-DNA complex and preventing the re-ligation of the DNA strand, leading to the formation of lethal DNA breaks. Following the initial response to irinotecan, secondary resistance is acquired relatively rapidly, compromising its efficacy. There are several mechanisms contributing to the resistance, which affect the irinotecan metabolism or the target protein. In addition, we have demonstrated a major resistance mechanism associated with the elimination of hundreds of thousands of Top1 binding sites on DNA that can arise from the repair of prior Top1-dependent DNA cleavages. Here, we outline the major mechanisms of irinotecan resistance and highlight recent advancements in the field. We discuss the impact of resistance mechanisms on clinical outcomes and the potential strategies to overcome resistance to irinotecan. The elucidation of the underlying mechanisms of irinotecan resistance can provide valuable insights for the development of effective therapeutic strategies.

## 1. Introduction

Irinotecan (CPT-11) is a chemotherapeutic agent that causes cancer cell killing by poisoning topoisomerase I (Top1) in the cell. It is a semisynthetic analog of camptothecin, which was originally isolated from the Chinese/Tibetan ornamental tree *Camptotheca acuminata* [1,2,3]. Resistance model-based studies uncovered several mechanisms of cellular resistance to this agent and, accordingly, multiple approaches were tested in clinical trials to circumvent the resistance [4,5,6]. Beside primary resistance, the major clinical problem is development of secondary resistance in the course of the drug treatment, which is often observed with DNA-damaging chemotherapeutic drugs, such as irinotecan [7,8,9] or doxorubicin [6,10,11]. It is commonly understood that the development of drug resistance in cancer cells is defined by the change in expression or function of the target protein [8,12], changes in the ability of cells to undergo cell death [13,14,15], or changes in drug metabolism [16,17,18,19].

Importantly, both in experimental and clinical settings, secondary resistance usually develops gradually in the course of multiple exposures to the drug. It is unclear why changes in transcriptome [9,12,16,18,19,20,21], metabolism [12,22,23], or mutations in drug targets [7,12,24,25,26,27,28,29,30] do not appear abruptly but require multiple drug administrations [12,14,16,31,32]. This puzzling requirement suggests a distinct mode of resistance that may require a gradual accumulation of a large number of mutations or epigenetic events. Each of these events may have a minor effect on the drug response, but cumulatively they provide significant resistance. Here, we discuss novel evidence for the existence of such a mechanism in the development of resistance to irinotecan.

## 2. Irinotecan’s Mode of Action

One of the classes of drugs that are frequently used in cancer therapy is inhibitors of DNA supercoil relaxing topoisomerases, including Top1 and Top2. Type I topoisomerases (Top1) cause relaxation of super helical DNA by generating a transient single-strand nick, followed by DNA relaxation and re-ligation. The enzyme subtypes perform very specialized functions, e.g., Class IA can only relax negative supercoiled DNA, whereas Class IB can introduce positive supercoils, relaxing and separating DNA molecules in daughter chromosomes after DNA replication. Top1 plays major role in transcription and replication and is highly active in pericentromeric and centromeric regions of the genome [33,34,35,36,37,38,39]. In contrast, type II topoisomerases (Top2) mediate ATP-dependent cleavage of both strands of the DNA double helix, followed by crossing of the DNA double-strand (ds) through the transiently opened gap [40]. Mammalian cells have two Top2 isoenzymes, Top2α and Top2β [35]. Top2α is associated with DNA replication and, as a consequence, cell proliferation, while Top2β may play a role in transcription [39,41].

Major Top1 or Top2 inhibitors function by stabilizing the transient complexes formed between these enzymes and DNA [37,42]. Stabilization of these otherwise fleeting “cleavable complexes” can lead to formation of double-strand breaks (DSBs) when the DNA replication forks collide with the topoisomerase-DNA complexes [37,42]. Similarly, the inhibitor-stabilized Top1 or Top2 “cleavable complexes” located on the transcribed template strand can also result in the formation of DSBs when RNA polymerase molecules collide with them within the transcribed DNA regions [40]. DSBs are recognized by the cell as lethal lesions and can trigger apoptosis [43,44]. These cytotoxic effects are responsible for the anti-cancer activity of topoisomerase inhibitors.

A most widely used Top1 inhibitor, irinotecan is a prodrug that, upon activation, generates an active compound, SN-38 [2,45,46,47,48]. Upon treatment of cancer cells, SN-38 binds to Top1 and stabilizes Top1-DNA complexes [2,49] (Figure 1). The SN-38 molecule stacks against the base pairs flanking the Top1-induced cleavage site and poisons the enzyme [2,49]. This conversion suppresses the 3′-OH free end and makes it unavailable for re-ligation. Inhibition of re-ligation of nicked DNA strands leads to single strand breaks, which have a high probability of conversion to highly toxic DSB. As the drug works not only during transcription [50], but also during replication, these DNA-Top1-SN38 covalent adjuncts could lead to replication fork stalling and the arrest of DNA replication.

High-intensity transcription and replication enhances the supercoiling of DNA to levels that can impede or halt these processes. As a potent transcription amplifier and replication accelerator, the proto-oncogene MYC must manage this interfering torsional change. In a recent study, a direct association of MYC with Top1 and Top2 was demonstrated [51]. Beyond recruiting topoisomerases, MYC directly stimulated their activities. These MYC complexes with Top1 and Top2 increased their activities at promoters, coding regions, and enhancers [51]. Such enhancement of the activity of Top1 and its recruitment to DNA may create additional cleavage sites upon the drug treatment. In line with this suggestion, it was demonstrated that the overexpression of MYC enhances the sensitivity of colon cancer cells to the parental drug, camptothecin [51,52]. At the same time, MYC can activate the DNA damage response, which results in induction of the DSB repair system [51]. This in turn could reduce the response to irinotecan, since DSB repair plays a critical role in survival of cells treated with irinotecan.

## 3. Clinical Use of Irinotecan

In the USA, irinotecan has been approved for use against colorectal cancer in combination with 5-fluorouracil (5-FU) and leucovorin (FOLFIRI regimen). With therapy regimens like FOLFIRI, the median survival rate of a patient with metastatic colorectal cancer has improved from 8 months to 24 months [47]. Irinotecan is also used in combination with Capecitabine (pro-drug of 5-FU) (XELIRI regimen). Currently, both regimens are considered first-line therapy for cancer treatment. Several studies have been conducted to assess the effects of these combinations, and they have demonstrated that they are equally effective, with certain variations in median survival rates [1,53,54,55,56,57,58,59]. The usage of these schemes is defined by several factors such as geographical regions, patient genetics, individual response rate, oncologist’s preference, and socio-economic factors.

Another promising combination of irinotecan is with antibodies against EGFR, such as Cetuximab, for treatment of patients with wild-type K-Ras colorectal cancers and certain other cancer types, with 5-FU/leucovorin as a first-line treatment [60]. It is important to note that initial studies suggested that patients with colorectal tumors characterized by high microsatellite instability (MSI) might respond better to irinotecan-based chemotherapy [61,62]. However, subsequent data did not support a predictive value of MSI status in relation to treatment response [63,64].

## 4. Irinotecan Analogs and Derivatives with Improved Properties

Beside irinotecan, there are several camptothecin derivatives poised to become pharmacological agents in various cancer treatment protocols. Among them, exatecan shows a great promise; it is 6 times more active than SN-38 and 28 times more active than topotecan [65,66]. In a recent study, the therapeutic effects of exatecan mesylate in its salt form (DX-8951f) were compared to effects of other FDA-approved camptothecin derivatives, such as irinotecan and topotecan, using human tumor xenografts in nude mice [67,68]. A total of sixteen human cancer lines were examined, of which six were colon cancer. Under the treatment conditions, the tumor growth inhibition rate (IR) of exatecan was superior to that of irinotecan or topotecan [66]. Importantly, irinotecan could be rendered ineffective due to the multidrug resistance pump-mediated efflux [68,69,70,71,72,73,74]. Exatecan, however, may overcome this resistance because it has a low affinity to MDR transporters [66]. Despite these favorable characteristics, exatecan remains in phase III clinical trials and has not yet been clinically approved because of its significant myelotoxicity [75]. Furthermore, as with other Top1 inhibitors, exatecan demonstrates a gastrointestinal and bone marrow toxicity [75]. Accordingly, an exatecan derivative, deruxtecan (Dxd), has been developed, which represents a conjugate with trastuzumab, a Her2-targeting antibody. Having similar inhibitory effects, deruxtecan has lower myelotoxicity than exatecan [76]. It has a greater safety, and has been approved by the FDA for the treatment of breast cancer or gastric or gastroesophageal adenocarcinoma [76,77].

Dxd and other developed conjugates of irinotecan derivatives with antibodies (TOP1-ADC) provide a series of advantages. Monoclonal antibody-mediated active targeting offers both selectivity and extremely high affinity because of specific antibody–antigen binding, which can distinguish tumor versus healthy cells based on antigen expression levels [66,78]. Consequently, TOP1-ADC is an effective approach to enhance the anti-tumor activity of both the monoclonal antibody and the Top1 inhibitor. Moreover, conjugation of the cytotoxic agent to the large, hydrophilic antibody restricts the penetration of the cytotoxic compound across the cellular membranes of antigen-negative normal cells, further lessening off-target side effects [65,79,80,81].

## 5. Irinotecan Activation and Detoxification

Upon administration through intravenous injection, a fraction of irinotecan is converted in the blood to active form SN-38 by carboxylesterases, which could lead to circulatory or digestive complications [82]. Nevertheless, the major portion of the drug reaches the liver for detoxification. Irinotecan uptake and transport into the liver is facilitated by multiple pumps, including OATP1B1 (SLCO1B1), ABCB1, MRP1 (ABCC1), MRP2 (ABCC2), and MXR (ABCG2) [83,84]. In the liver, inactive form irinotecan is converted to an active form SN-38 by the low affinity carboxylesterases CES1 and CES2 with a relatively low efficiency (<3%) [27,85]. In an alternative pathway, irinotecan could be oxidized, resulting in the inactive metabolites APC (7-ethyl-10-[4-N-(5-aminopentanoic acid)-1-piperidino] carbonyloxycamptothecin) and NPC (7-ethyl-10-[4-(1-piperidino)-1-amino] carbonyloxycamptothecin) (Figure 2). NPC can also be metabolized into active SN-38 by CES1 and CES2. Further, SN-38 could undergo glucuronide conjugation by UGT1A1 (UDP-glucuronosyltransferase) and enter the detoxification pathway (Figure 2) [83,85,86]. SN-38 glucuronide (SN-38G) is excreted in the gut and is hydrolyzed by β-glucuronidase. Alternatively, SN-38 can be oxidized and inactivated by the P450 CYP3A. Components that are produced from irinotecan metabolism are highly pH sensitive, and hence, at any point these metabolites could become active or inactive [87]. Irinotecan resistance can be attributed to one or many intermediate points in this metabolic pathway, as discussed below.

## 6. Systems That Control Irinotecan Resistance

### 6.1. Glucuronidation and Irinotecan Resistance

High sensitivity to irinotecan associates with genetic variance in the UGT1A, CYP3A, and ABC gene families which play a major role in the metabolism of the drug [23,49,88]. Since SN-38 is glucuronidated by a family of UGT1A enzymes [28], colon, lung, and breast cancer cells that have high expression levels of UGT1A demonstrate resistance to SN-38. Furthermore, meta-data-based pharmacogenetic study of colorectal cancer patients established strong associations between germline variants of *UGT1* and clinical outcomes, and have strongly supported the role of the UGT1A enzyme pathway in mediating the response to SN-38 [89,90]. The best example pertains to the clinically actionable marker *UGT1A1*28*, which is associated with reduced UGT1A1 activity and irinotecan-induced severe neutropenia [16,90] (Figure 2). Subsequently, in 2005, the US Food and Drug Administration (FDA) issued a recommendation for *UGT1A1*28* testing of potential irinotecan users [91].

Members of the drug transporting system responsible for irinotecan influx from blood to hepatocytes, such as SLCO1B1, may also be involved in severe toxicity to irinotecan (Figure 2). The presence of a genetic variant of SLCO1B1 (521T > C) was correlated with higher toxicity in metastatic colorectal cancer patients treated with irinotecan as the first line of treatment [47,49,92]. It was hypothesized that the 521T > C variant somehow alters the membrane permeability of the drug and enhances the transport of the drug inside the cells [92].

### 6.2. Cytochrome P450 and Irinotecan Resistance

Another critical pathway that orchestrates response to irinotecan is P450 enzymes. Irinotecan is metabolized by intrahepatic cytochromes P450, i.e., CYP3A4 and CYP3A5, into inactive metabolites—APC and NPC [48,93] (Figure 2). In contrast to APC, NPC can be converted to SN-38 by CES1 and CES2 in the liver [94]. Further, the CYP3A5*3 polymorphism, which leads to reduced enzyme activity, has been associated with significantly longer progression-free survival in patients with metastatic colorectal cancer (CRC) [95,96]. Furthermore, there was a statistically significant correlation between CYP3A5 expression and tumor response to irinotecan therapy, suggesting a tumor-autonomous resistance to the treatment through increased CYP3A5-mediated metabolism [95].

### 6.3. MDR Pumps and Irinotecan Resistance

A very important factor that defines the development of resistance is ATP-binding cassette (ABC) transporters, such as the multi-drug resistance 1 P-glycoprotein gene (ABCB1) and the multi-drug resistance-associated protein 2 gene (ABCC2) (Figure 2), which facilitate the efflux of irinotecan and its metabolites [91,97]. Overexpression of ABCG2 is even more significant in irinotecan resistance [21,73,98]. Cell-based assays showed that higher resistance was observed for SN-38 (approximately 50-fold), irinotecan (17–48-fold), and topotecan (approximately 40-fold) in ABCG2-overexpressing cells compared to ABCB1-overexpressing cells [32,99,100]. To counter the resistance provided by the ABCG2 transporter, SN38-loaded pegylated (polyethylene glycol) PLGA [poly(lactic-co-glycolic acid)]-verapamil nanoparticles (NPs) were developed. Because of the verapamil component, the compound inhibits MDR pumps and, in addition, it reduces expression of ABCG2 [21,73,98,98,101]. In conclusion, sufficient uptake of SN38-PEG-PLGA-Ver NPs and a significant decrease in expression of ABCG2 were achieved, indicating a successful approach towards counteracting resistance associated with drug efflux.

### 6.4. Xenobiotic Receptors and Irinotecan Resistance

The xenobiotic receptors PXR/SXR and the retinoic receptor RXR that regulate expression of UGT1A1, CYP3A4 and ABC transporters significantly contribute to irinotecan resistance [102,103,104]. For example, cDNA-mediated expression of PXR in cultured colon cancer cells LS174T made the cells resistant to SN-38 by enhancing the glucuronidation of SN-38 to SN-38G [103,105]. Conversely, shRNA-mediated downregulation of PXR decreased the SN-38G/SN-38 ratios, in accordance with UGT1A1 downregulation [105]. In a feedback loop, SN-38 activates PXR in human colon cancer cell lines and induces CYP3A4, CYP3A5, UGT1A1, and the ABC transporter, ABCC2 [103,105]. This resistance was reversed by PXR repression [17]. Similarly, endogenous SXR is activated in response to SN-38 [71] and binds to the promoter of the CYP3A4 gene to induce its expression. Further using ChIP, it was demonstrated that SXR translocates into the nucleus and associates with RXR upon SN-38 treatment. RNA interference experiments confirmed SXR involvement in CYP3A4 expression and identified CYP3A5 and ABCC2 as SXR target genes. Consequently, cells overexpressing SXR show reduced sensitivity to irinotecan treatment [71]. Therefore, activation of PXR/SXR results in induction of various drug-metabolizing enzymes and drug transporters, all of which play a role in the disposition and excretion of irinotecan, SN-38, and SN-38G, though detailed mechanisms of this regulation remain unclear.

### 6.5. Cancer Cell Stemness and Irinotecan Resistance

An important factor in resistance to irinotecan-containing therapies is cancer cell stemness. Cancer stem cells (CSC) have self-renewing capabilities due to the expression of a series of transcription factors, such as NANOG, OCT4, or SOX2. Generally, CSC demonstrate drug resistance due to a number of properties, including low rate of division, thus spending the majority of time in G0 [106], high levels of expression of cytochrome p450 isoforms [48], high levels of MDR pumps [21,48,107], and other factors.

Indeed, expression of CD133, a marker for stemness of CRC, was found to correlate with the resistance. Such CD133+ tumorigenic cells in colon cancer represent approximately 2.5% of the total tumor cells [108]. These cells are able to initiate tumor formation in mice and spheroids in 3D culture. These cells demonstrated high resistance to irinotecan [109,110]. Furthermore, losing expression of CD133 in adherent culture correlated with the loss of the resistance [109,110]. One factor of the resistance was probably high expression of CYP3A4 [18]. Another possible contributor to the resistance was elevated levels of ABC transporters [111], coupled with an increased mitochondrial ATP output. For example, ABCG2 is overexpressed in colon CSCs [112], and was reported to be a major driver of drug resistance of these cells. Furthermore, overexpression of ABCG2 could potentially be a general marker of CSC [98,101,113,114].

### 6.6. Autophagy and Irinotecan Response

DNA breaks following irinotecan treatment can activate apoptosis via p53 and other mechanisms [43,53,115]. In parallel, they activate a protective autophagy that has a tumor promoter effect, since it provides tumor cells with energy and amino acids required for survival and proliferation [116,117]. Activation of autophagy by irinotecan involves the JNK- and p38-MAPK signaling pathways and generation of reactive oxygen species (ROS) [117,118]. Autophagy was identified as an important mechanism of resistance to irinotecan treatment in cancers and cancer cells [116,117,119]. Clinical data further support the role of autophagy in the irinotecan resistance. Indeed, in CRC patients undergoing irinotecan chemotherapy, overexpression of autophagy-associated proteins, such as Beclin-1, associates with low survival [116,118,120,121,122]. Furthermore, a series of preclinical studies demonstrated that autophagy inhibitors, such as chloroquine/3-methyladenine, can reverse CRC resistance to irinotecan [117,122,123,124]. Moreover, in a recent study, toosendanin (TSN) that blocks autophagy flux could sensitize triple-negative breast cancer cells to SN-38/irinotecan-induced cytotoxicity both in vitro and in vivo [125]. Altogether these findings suggest that inhibition of autophagy may become an interesting approach towards improving the response to irinotecan therapy.

## 7. Mutations That Change Top1 Conformation and Function

Expression of the irinotecan target Top1 was reported to be a predictive biomarker for irinotecan sensitivity [34,126]. Furthermore, resistance to Top1-targeting drugs can be developed via mutations that affect Top1 structure or expression [7,8,24].

Top1 has four domains: the NH2-terminal domain (residues 1–214), the core domain (residues 215–635), which can be divided into subdomains I, II, and III, the linker domain (residues 636–712), and the COOH-terminal domain (residues 713–765). The poorly conserved linker domain, which connects the core and COOH-terminal domains, is highly flexible [127] and is dispensable for the catalytic activity [12]. Nevertheless, the lack of a functional linker associates with reduced sensitivity to irinotecan [127,128,129]. It was proposed that increasing flexibility of the linker domain alters the conformation imposed by the drug binding, giving rise to the irinotecan-resistant enzyme [127,128,129].

A number of point mutations in Top1 that confer resistance to SN38 have been identified in tumor cell culture (e.g., p.F361S, p. R364H, p.E418L, p.G503S, p.D533G, p.A653P, and p.N722S) (Figure 3). However, these mutations have not been seen in clinical samples. This apparent contradiction is probably related to mutants’ reduced growth rate [24,128,129,130]. Accordingly, in a heterogenic tumoral population where there is ample diversity among the clones, a fraction of Top1 mutation-carrying clones could be low and, therefore, not picked up in mutation analysis [24,130]. In addition to point mutations in Top1, copy number variations of the Top1 gene that associate with resistance were also reported in different cultured cancer cells [131].

## 8. Novel Mechanisms of Drug Resistance to SN-38 Involve Mutations at Top1 Binding Sites That Reduce Top1-Mediated DNA Cleavage [36]

In a recent development, we studied a puzzling need of multiple exposures to drugs in selection of resistant cancer cell mutants in culture. Our investigation involved a cloned, genetically identical population of HCT116 colon cancer cells. This model could recapitulate clinical conditions in clonal emergence and evolution of cancers. Multiple exposures of the genetically identical population to irinotecan in a dose escalation setting led to the evolution of drug resistance. Our initial expectation was that conventional resistance mechanisms would emerge, such as Top1 mutations or alterations in the transcriptome landscape that could promote cell survival. To elucidate these mechanisms, we independently selected three clones and conducted whole genome sequencing and transcriptome analysis, comparing them to the parental irinotecan-sensitive clone. Surprisingly, RNAseq did not demonstrate any changes in gene transcription related to drug metabolism, detoxification mechanisms, or MDR pumps, indicating that alterations in the transcriptome were unlikely to be involved in the development of resistance.

Importantly, we detected hundreds of thousands of SNPs and InDels in the resistant mutants compared to the parental clone. Among them, we did not find mutations that affect Top1 or genes that belong to pathways associated with either cell survival or DNA repair. Therefore, it is unlikely that changes in gene function or expression are involved in this adaptation process (Figure 4).

Strikingly, 15% of the mutations in the three independent isolates were identical. This large number of common mutations clearly indicates a non-random mechanism for their generation. Metadata analysis indicates that the mutations occurred at Top1 binding/cleavage sites in the genome. Interestingly, the majority of these sites corresponded to various types of satellites, including alpha satellites and satellite-II, indicating that Top1 preferentially interacts with and cleaves DNA at satellite sites. In the parental clone of the HCT116 cells, we observed a significantly higher number of these sites than in the reference genome that represents an average of several normal human genomes. Considering that the HCT116 line demonstrates satellite instability, it appears that this instability generates additional Top1 binding/cleavage sites in the process of cancer evolution. This increased number of Top1 sites may significantly contribute to the elevated sensitivity of these cells to irinotecan.

Importantly, mutations generated in the irinotecan resistance mutants carried signatures of the double strand breaks repair mechanisms (HR: Homologous recombination and NHEJ: Non-homologous end joining). Accordingly, considering their locations at the Top1-binding sites, these mutations must result from the repair of DSB created by Top1 upon SN-38 treatments.

The most striking finding was that in the case of heterozygocity in the parental clone: where one allele was normal (found in the reference genome) and another one mutated due to cancer evolution, the latter allele was lost preferentially upon development of irinotecan resistance. Such loss of heterozygocity can only result from the HR repair of DSB. The disproportionately high loss of the “cancer allele” indicated that Top1-generated DSB took place in this allele preferentially, and it was repaired by copying of the reference genome sequence from the homologous chromosome. Therefore, reversion to the normal reference genome allele upon treatment with SN-38 must eliminate the Top1 cleavage/binding site at this place. Accordingly, we hypothesized that upon initial treatments with SN-38, Top1 that associates with specific binding sites on the chromosomes creates single strand breaks that can be converted to toxic DSB at these sites. Repair of these DSBs via HR (and possibly NHEJ) creates mutations at these sites that prevent binding of Top1 to them upon following exposures to the drug. Indeed, in the resistant mutants, we observed a significantly reduced association of Top1 with the genome. Furthermore, upon exposures of the resistant mutants to SN-38, we observed dramatically fewer DSB, as monitored by formation of the γH2AX foci [36].

Taken together, these findings suggest that the primary pathway for adaptation to SN-38 is not linked to functional gene mutations, such as Top-1, or alterations in gene expression. Rather, mutations are generated after each exposure to the drug, which effectively “close” Top1 binding/cleavage sites on the chromosomes. These mutations gradually accumulate upon multiple exposures and ultimately render cells resistant to the drug. In other words, the development of irinotecan resistance is an inherent feature of the drug’s mechanism of action, where resistant mutations are directly produced from the repair of DSBs caused by Top1. Due to this intrinsic mechanism, resistance can be acquired rapidly, within just a few cycles of drug administration.

These findings suggest interesting strategies for the suppression of the development of irinotecan resistance. Indeed, the uncovered mechanism of the resistance development is dependent on the efficiency of the DSB repair. According to this analysis, HR repair is the major factor in generating resistance mutations, but NHEJ may also be actively involved in their generation. Therefore, the model predicts that inhibiting HR alone or in combination with NHEJ would not only increase the sensitivity of cancer cells to irinotecan, but also prevent the development of secondary resistance.

Another interesting conclusion from this mechanism is that, since the number of Top1 sites is gradually reduced upon development of the resistance, Top1’s ability to relax DNA supercoils should gradually get reduced. Since relaxation of DNA supercoils is essential for replication and transcription, the DNA relaxation activity of Top1 must be taken over by another enzyme, most likely Top2. Accordingly, development of resistance to irinotecan may enhance sensitivity of cells to Top2 inhibitors, such as doxorubicin. This possibility may guide the patient treatment strategy.

## Figures and Tables

**Figure 1 ijms-24-07233-f001:**
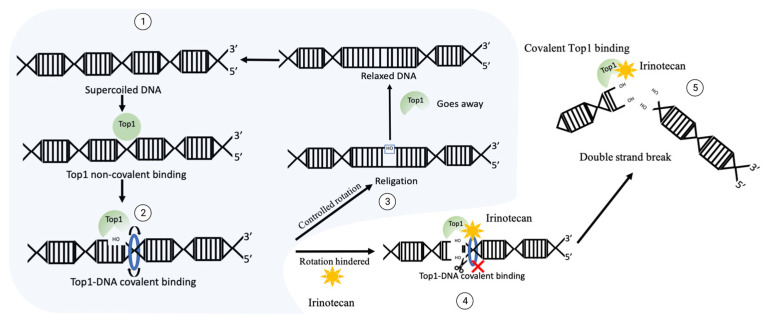
The mechanism of DNA generation breaks upon irinotecan exposure. Normal function of Top1: ① supercoiled sensing and nick development by Top1 leads to ② relaxation, then ③ Top1 re-ligates the nicked backbone and leaves the site. In the case of irinotecan exposure, Top1 gets covalently crosslinked to DNA, then ④ ligation of 3′-OH free end is blocked, which could translate into ⑤ double strand break and facilitate cytotoxicity.

**Figure 2 ijms-24-07233-f002:**
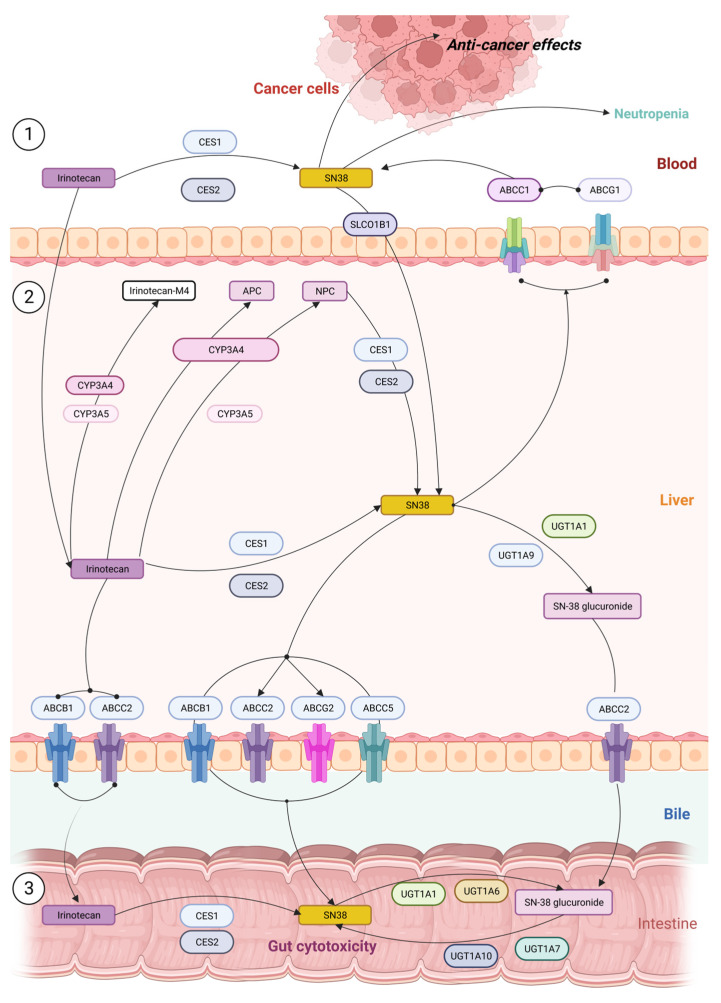
Pathways of irinotecan metabolism in the liver cells ① showing the conversion of inactive form to active SN-38 leading to neutropenia; ② the drug enters via ABC transporters where it is converted to derivatives irinotecan M4, APC, and NPC; ③ the excreted version of irinotecan or SN38G is reconverted to active form SN-38 in intestine and causes GI toxicity.

**Figure 3 ijms-24-07233-f003:**
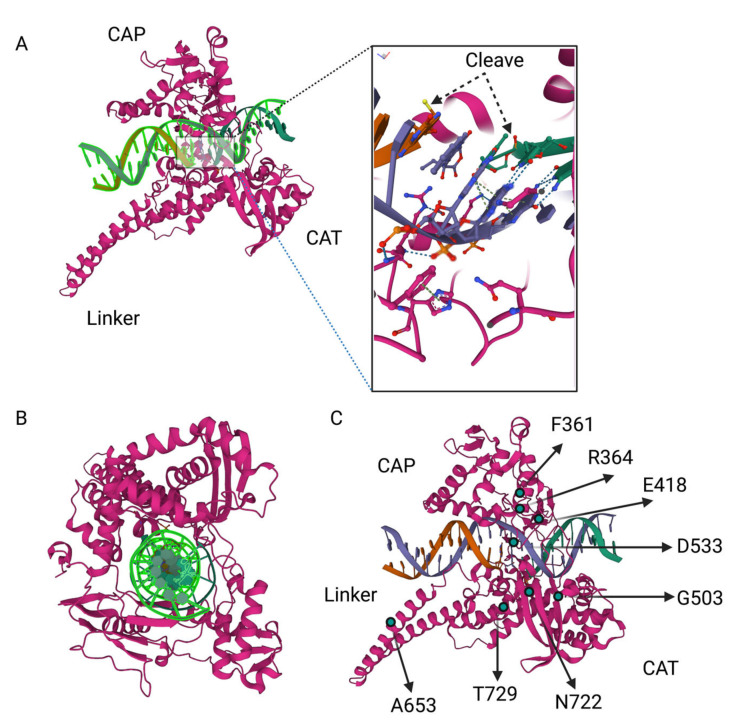
Crystal structure model for topoisomerase I. (**A**) Showing two domains, DNA placement and cleavage, (**B**) sections showing Top1 wrapped around DNA, (**C**) Irinotecan-resistance mutations in Top1 (not exact scale and is for representation based on data from public depository, http://doi.org/10.2210/pdb1k4t/pdb, accessed on 12 February 2023).

**Figure 4 ijms-24-07233-f004:**
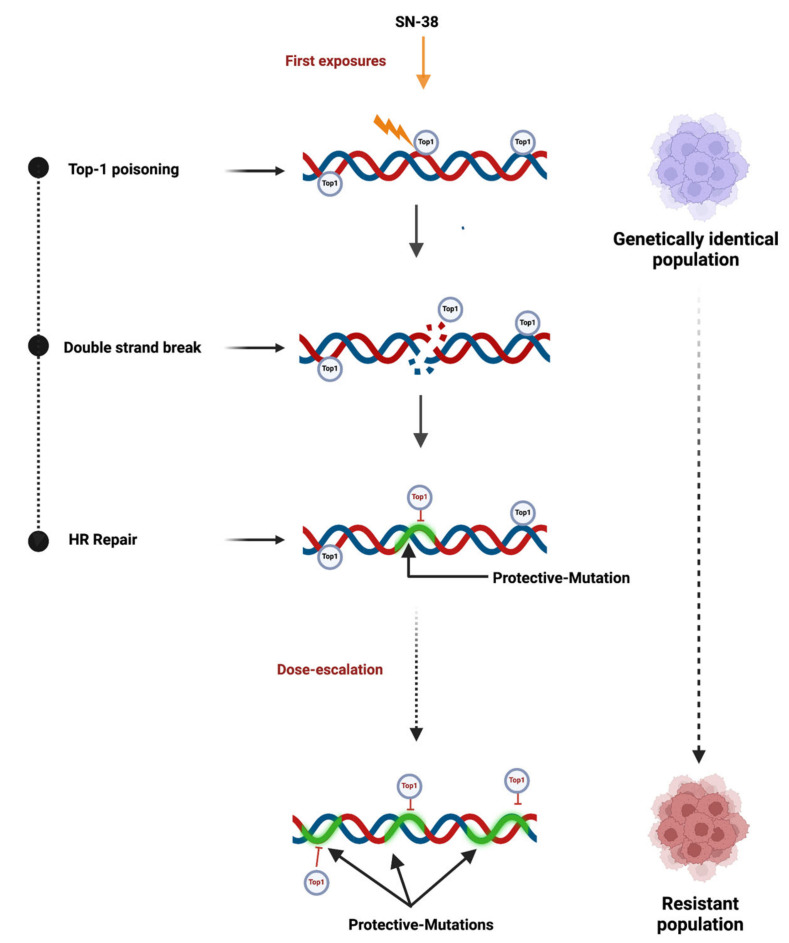
Study model for dose escalation-mediated drug resistance. Resistance mutations result from cycles of creating DSB and their repair at the Top1-binding sites.

## Data Availability

Not applicable.

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
