# Peer review of "Resistance to TOP-1 Inhibitors: Good Old Drugs Still Can Surprise Us"

_ijms, 2023, doi:10.3390/ijms24087233_

Round 1

Reviewer 1 Report

This is a review article about resistance to irinotecan (SN-38). A review on irinotecan (and other TOP1-targeting drugs) should be very helpful, as these widely used drugs no longer enjoy the literature attention that they used to have. This MS contains a helpful overview of what is known about the mechanism of action of irinotecan and the resistance to this drug. Much of the review, however, is devoted to the authors' own study that appears to have delineated a new and scientifically interesting mechanism of irinotecan resistance. This could have been appropriate once the underlying study is published. However, that study (ref. 35) is merely a preprint, and I don't feel that it's right to publish a review article that is largely devoted to a paper that has not yet made it through peer review.

In addition, this MS is not well organized and is not adequately edited. It would be logical to start from the mechanism of the drug's action, discuss any relevant drug derivatives, summarize the clinical experience and then proceed to the mechanisms of resistance. However, this sequence is not followed: for example, why is a section about autophagy (a mechanism of resistance) placed  before the clinical experience and the drug analogs? Serious editing is needed, it even seems that a non-final MS may have been accidentally submitted. For example, lines 54-68 duplicate the text above them. The authors need to do a lot of language editing: there are missing verbs, sequence of tenses, singular/plural mix-ups; the use of articles need to be fixed. 

For these reasons, I think that this potentially helpful review is not ready to be considered for publication. 

Author Response

We appreciate the reviewer’s comments and feedback on our review article and would like to address the criticism as attached.

Reviewer 2 Report

Dear authors,

This review was well-written and has many apparent and instructive figures. I would like to suggest the inclusion of some new references that are important in irinotecan's research:

- Pharmacol Res . 2019 Oct;148:104398. doi: 10.1016/j.phrs.2019.104398.

  - Biochem Pharmacol . 2017 Dec 15;146:53-62. doi: 10.1016/j.bcp.2017.10.003.   - Drugs . 1996 Oct;52(4):606-23. doi: 10.2165/00003495-199652040-00013.  

Author Response

(The authors gave the same response as above.)

Reviewer 3 Report

This review discusses many relevant aspects of irinotecan resistance mechanisms.  It is comprehensive and should be of interest given the recent success of ADC of newer irinotecan analogs in clinical trials.  I have the following suggestions for revision before the review is published.

 1.       Line 41:54: One of the classes of drug that are frequently used as cancer therapies are inhibitors of DNA supercoil relaxing Topoisomerases including Top1 and Top2….. Exact paragraph is repeated starting from Line 54.

2.       Line 44: Prokaryotic topoisomerase I (class IA) can only relax negative supercoiled DNA, whereas eukaryotic topoisomerase I (class IB) can introduce positive supercoils, separating the DNA of daughter chromosomes after DNA replication, and relax DNA. This is not accurate.  There are many eukaryotic class IA topoisomerases, including human Top3A and Top3B that only relax negative supercoiled DNA. Instead of saying “Prokaryotic topoisomerase I (class IA)”, should state “Class IA topoisomerases can only relax negative supercoiled DNA…”

 3.       It should be stated more clearly in line 41 and 69 that topoisomerase inhibitors used in cancer therapy and have mechanism involving stabilization of otherwise transient (“cleavable”) complexes formed between Top1 or Top2 and DNA are topoisomerase poisons. Some topoisomerase inhibitors that are not poisons inhibit the activity by acting at steps other than DNA ligation.

 4.       The authors mentioned on page 7 very interesting new findings on a novel mechanism of resistance “that upon initial treatments with SN-38, Top1 that associates with specific binding sites on the chromosomes creates single strand breaks that can be converted to toxic DSB at these sites. Repair of these DSBs via HR (and possibly 330 NHEJ) creates mutations at these sites that prevent binding of Top1 to them upon following exposures to the drug.” It may not be appropriate to go into such details of these unpublished results especially since there is no mention on where the readers can find the supporting data such as “upon exposures of the resistant mutants to SN-38, we observed dramatically fewer DSB, as monitored by formation of the É£H2AX foci.” It would be preferable to have availability of the new results in a public database or a biorxiv paper (which can be cited as a reference).

 5.       Starting from line 354, authors proposed that “Since relaxation of DNA supercoils is esential (typo) for replication and transcription, the DNA relaxation activity of Top1 must be taken over by another enzyme, most likely Top2. Accordingly, development of resistance to irinotecan may enhance sensitivity of cells to Top2 inhibitors, such as doxorubicine (typo). This  possibility may guide the patient treatment strategy.” It should be noted that among the two Top2 enzymes in human, Top2B provides a more important contribution to relaxation during transcription.  The anticancer effect of doxorubicin is due to it acting as a topoisomerase poison on Top2A, which is known to provide the decatenation activity during replication termination.  The trapping of Top2B cleavage complex by doxorubicin results in undesirable toxicity on non-cancer cells, causing chromosomal translocation. 

 6.       For Figure 3, the authors should provide the code for the PDB file used to generate the figures of the TOP1 structures shown here.

Author Response

(The authors gave the same response as above.)

Round 2

Reviewer 1 Report

The revised paper has been improved but I may be too old-fashioned and I don't feel that it's proper to describe a study that has not been peer-reviewed in a review article. I would wait until Ref. 36 has been accepted for publication. 

Author Response

The reviewers do not have any request for improvement of the manuscript and also have agreed that we addressed their criticism, and quality of the manuscript has significantly improved. Therefore, we do not have anything to answer the reviewer. Hence this is now in Editor's hands to decide regarding the publication.